# Non-Contrast-Enhanced Multiparametric MRI of the Hypoxic Tumor Microenvironment Allows Molecular Subtyping of Breast Cancer: A Pilot Study

**DOI:** 10.3390/cancers16020375

**Published:** 2024-01-16

**Authors:** Silvester J. Bartsch, Klára Brožová, Viktoria Ehret, Joachim Friske, Christoph Fürböck, Lukas Kenner, Daniela Laimer-Gruber, Thomas H. Helbich, Katja Pinker

**Affiliations:** 1Department of Biomedical Imaging and Image-Guided Therapy, Division of Structural and Molecular Preclinical Imaging, Medical University of Vienna, 1090 Vienna, Austria; 2Department of Experimental and Laboratory Animal Pathology, Clinical Institute of Pathology, Medical University of Vienna, 1090 Vienna, Austria; 3Unit of Laboratory Animal Pathology, University of Veterinary Medicine Vienna, 1210 Vienna, Austria; 4Department of Internal Medicine III, Division of Endocrinology and Metabolism, Medical University of Vienna, 1090 Vienna, Austria; 5Computational Imaging Research Laboratory, Department of Biomedical Imaging and Image-Guided Therapy, Medical University of Vienna, 1090 Vienna, Austria; 6Comprehensive Cancer Center, Medical University Vienna, 1090 Vienna, Austria; 7Christian Doppler Laboratory for Applied Metabolomics, Medical University Vienna, 1090 Vienna, Austria; 8Center for Biomarker Research in Medicine (CBmed), 8010 Graz, Austria; 9Breast Imaging Service, Department of Radiology, Memorial Sloan Kettering Cancer Center, New York, NY 10065, USA

**Keywords:** breast-cancer characterization, hypoxia-induced angiogenesis, hyperoxic BOLD MRI, IVIM MRI, intrinsic-contrast MRI, preclinical imaging

## Abstract

**Simple Summary:**

Breast cancer, the second-leading cause of mortality among women worldwide, is often diagnosed through invasive tissue sampling. This method may not fully represent the tumor’s overall physiology, potentially leading to suboptimal treatment strategies and tumor recurrence. A non-invasive approach using MRI to quantify the hypoxic environment within the tumor could significantly enhance the accuracy of breast-cancer-subtype prediction. Our study introduces non-invasive imaging markers for hypoxia and related angiogenesis, utilizing hyperoxic-blood-oxygen-level-dependent (BOLD)-MRI and intravoxel-incoherent-motion (IVIM)-MRI techniques. These markers correlate with microvessel density and maturity as assessed by histology, enabling the distinction between the less aggressive luminal A and the more aggressive triple-negative breast cancers.

**Abstract:**

Tumor neoangiogenesis is an important hallmark of cancer progression, triggered by alternating selective pressures from the hypoxic tumor microenvironment. Non-invasive, non-contrast-enhanced multiparametric MRI combining blood-oxygen-level-dependent (BOLD) MRI, which depicts blood oxygen saturation, and intravoxel-incoherent-motion (IVIM) MRI, which captures intravascular and extravascular diffusion, can provide insights into tumor oxygenation and neovascularization simultaneously. Our objective was to identify imaging markers that can predict hypoxia-induced angiogenesis and to validate our findings using multiplexed immunohistochemical analyses. We present an in vivo study involving 36 female athymic nude mice inoculated with luminal A, Her2+, and triple-negative breast cancer cells. We used a high-field 9.4-tesla MRI system for imaging and subsequently analyzed the tumors using multiplex immunohistochemistry for CD-31, PDGFR-β, and Hif1-α. We found that the hyperoxic-BOLD-MRI-derived parameter Δ*R2** discriminated luminal A from Her2+ and triple-negative breast cancers, while the IVIM-derived parameter *fIVIM* discriminated luminal A and Her2+ from triple-negative breast cancers. A comprehensive analysis using principal-component analysis of both multiparametric MRI- and mpIHC-derived data highlighted the differences between triple-negative and luminal A breast cancers. We conclude that multiparametric MRI combining hyperoxic BOLD MRI and IVIM MRI, without the need for contrast agents, offers promising non-invasive markers for evaluating hypoxia-induced angiogenesis.

## 1. Introduction

Breast cancer is recognized for its significant intratumoral heterogeneity, driven by high phenotypic plasticity and dynamic selective pressures from the hypoxic tumor microenvironment [1,2,3]. In 2022, phenotypic plasticity was identified as a central hallmark of cancer progression [4,5]. Despite this heterogeneity, treatment decisions often rely on invasive tissue sampling, which does not necessarily capture the biology of the entire tumor, therefore leading to inadequate treatment plans [6].

Hypoxia, or the insufficient supply of oxygen within certain tumor regions, is a key factor in the tumor microenvironment that drives the malignant potential of the tumor [7,8,9]. From the need to react to alternating selective pressures of the hypoxic tumor microenvironment, breast cancers invoke the co-adaptive angiogenic switch, characterized by the growth of microvessels to counteract hypoxic conditions [10]. Hypoxia-induced angiogenesis and increased microvessel density are recognized as crucial prognostic biomarkers in breast cancer [11,12]. The accumulation of hypoxia-inducible factor 1 (Hif1-α), a primary driver of angiogenesis through the activation of factors like vascular endothelial growth factors (VEGFs), is associated with tumor progression and the emergence of aggressive, treatment-resistant breast cancers of the triple-negative breast-cancer molecular subtype [13,14,15,16,17,18,19]. 

Efforts have been made to develop imaging markers of hypoxia-induced angiogenesis that are prognostic of the breast-cancer malignant potential and that allow the non-invasive characterization of breast cancers in its entirety [20,21]. Multiparametric MRI (mpMRI), utilizing various functional parameters, offers detailed insights into cancer hallmarks [21,22,23,24]. The development of novel mpMRI protocols using techniques that are independent of the application of gadolinium-based contrast agents (GBCA) may promote the use of non-contrast-enhanced mpMRI protocols for breast-cancer characterization.

In this study, we combined two non-invasive, non-contrast-enhanced MRI techniques, blood-oxygen-level-dependent (BOLD) MRI and intravoxel-incoherent-motion (IVIM) MRI, into an mpMRI protocol to investigate the potential of imaging parameters from both techniques to map the delivery of oxygen within breast cancers [25,26,27]. BOLD MRI is dependent on the ratio of oxygenized and deoxygenized hemoglobin, which induces susceptibility gradients and alters the transverse relaxation rate *R2** of nearby protons [28]. The dependency of BOLD-MRI signal on blood oxygenation is most frequently associated with brain functional MRI studies, where a signal change is associated with regional brain activity [29,30]. To introduce diagnostic variation into the measurement of tumor *R2**, so-called “hyperoxic gas challenges” are performed. The decrease in *R2** following a hyperoxic gas challenge with up to 100% oxygen in the breathing gas is proportional to the increased oxygenation of hemoglobin and therefore highlights tumor regions with functional blood oxygen delivery. Meanwhile, IVIM MRI, a diffusion-weighted-imaging (DWI) technique, distinguishes between extravascular and intravascular proton diffusion [31,32]. Routinely used DWI protocols, which include b-values between 300 s/mm^2^ and 1000 s/mm^2^, enable a quantification of the apparent diffusion coefficient, a surrogate marker of tissue cellularity, via the signal decrease due to random proton diffusion. Following the measurement of b-values lower than 300 s/mm^2^ in IVIM-MRI protocols, the non-random, microvascular diffusion contributes predominantly to the signal loss, and the perfusion-related IVIM fraction fIVIM and perfusion coefficient D* can be estimated. IVIM MRI has been used to differentiate malignant breast lesions from healthy tissue [33,34,35,36] and appears to provide promising imaging markers of angiogenesis in breast cancer.

Of note, the validation of in vivo imaging markers obtained through mpMRI can be enhanced by further comparing MRI parameter maps with fluorescent multiplexed immunohistochemistry (mpIHC). Unlike traditional singleplex immunohistochemistry, mpIHC can display up to ten histological markers on the same tissue section [37]. In our study, we therefore selected CD31 and PDGFR-β as microvasculature-specific mpIHC markers and Hif1-α as a hypoxia-specific mpIHC marker to validate our findings concerning in vivo imaging markers.

Our specific aim in this study was to identify prognostic imaging markers of hypoxia-induced angiogenesis in xenograft mouse models of three breast-cancer molecular subtypes and to compare these findings with mpIHC staining. More generally, we introduce a novel approach for non-contrast-enhanced mpMRI, sensitive to both the oxygen delivery to tumor subregions and hypoxia-induced neoangiogenesis. This approach may provide predictive insights into breast-cancer molecular subtypes.

## 2. Materials and Methods

### 2.1. Breast-Cancer Xenograft Model

This preclinical investigation received approval from the Austrian Federal Ministry of Education, Science and Research under the project number BMFWF-66.009/0284-WF/V/3b/2017. All procedures in this study adhered to the guidelines outlined in the European Community’s Council Directive of 22 September 2010 (2010/63/EU). Breast cancer cell lines representing different molecular subtypes, including luminal A (MCF-7), Her2+ (SKBR-3), and triple-negative (MDA-MB-231), were obtained from the American Type Culture Collection in Manassas, VA, USA. These cell lines were cultured in a controlled environment at 37 °C, 95% humidity, and 5% atmospheric CO2. The aggressiveness of these breast cancer cell lines was categorized as low (MCF-7, luminal A), medium (SKBR-3, Her2-neu positive non-luminal), or high (MDA-MB-231, triple-negative). MCF-7 cells were maintained in RPMI medium from Thermo Fisher Scientific, Waltham, MA, USA, while SKBR-3 and MDA-MB-231 cells were cultured in Dulbecco’s Modified Eagle Medium (DMEM) from the same supplier. All cell culture media were supplemented with 10% fetal bovine serum and 2% penicillin/streptomycin.

For the animal study, BALB/c mice (*n* = 36), aged approximately 4–6 weeks, were obtained from Charles River in Wilmington, MA, USA. Female mice were inoculated with 2 × 10^7^ breast cancer cells representing the three molecular subtypes (MCF-7, *n* = 16; SKBR-3, *n* = 10; MDA-MB-231, *n* = 10) into their flank. To support the growth of luminal A breast cancer cells in the mice, estrogen pellets (0.72 g/60-day release, Innovative Research of America, Sarasota, FL, USA) were implanted in the neck area. Tumors were allowed to grow for 1–2 weeks until they reached suitable volumes for mpMRI analysis, with a maximum tumor diameter of 1.0 cm.

Since some mpMRI data had to be excluded due to motion artifacts and since mpIHC data were not available for all mice due to processing issues, the sample sizes for BOLD MRI, IVIM MRI, and mpIHC varied.

### 2.2. Experimental Setup

All mpMRI examinations were performed using a 9.4-tesla Bruker BioSpec 94/30 USR system (Bruker, Ettlingen, Germany). Anesthesia, as well as vital parameter monitoring, were performed as described previously [27]. For oxygen-enhanced BOLD-MR images, the fraction of oxygen in the anesthetic gas was elevated to 100% using an air–oxygen blender (Sensor Medics Corporation, Yorba Lina, CA, USA). After the MRI experiments, mice were euthanized via cervical dislocation, and tumors were harvested for histological analyses.

### 2.3. mpMRI Protocol

T1 mapping was performed to obtain anatomical reference images of the breast cancer xenografts using a 2D RARE sequence with variable repetition times (VTR: 1472 ms, 2000 ms, 3000 ms, 4000 ms, 5000 ms, 7000 ms, 8000 ms, and 9000 ms; echo time (TE): 21.0 ms, matrix size: 128 × 128 pixels; number of slices: 10; slice thickness: 1 mm; spatial resolution: 0.278 × 0.222 mm; acquisition time: 16 min).

For BOLD MRI, T2* maps were acquired using a multi-gradient echo sequence (TR: 850 ms; TE: 14 echoes from 7–34.47 ms with an interval of 2.11 ms; matrix size: 128 × 128; number of slices: 10; slice thickness: 1 mm; spatial resolution: 0.278 × 0.222 mm; acquisition time for 3 cycles: 4 min). The measurement cycles were repeated three times for baseline BOLD acquisitions (using 21% oxygen) and challenged BOLD measurements (using 100% oxygen). Using the built-in image-sequence-analysis tool of the ParaVision 7 software suite (Bruker, Ettlingen, Germany), T2* maps were obtained.

IVIM-MRI measurements were conducted using a series of echo-planar-imaging (EPI)-based scan acquisitions (TR: 3200 ms; TE: 19.0 ms; matrix size: 81 × 154 pixels; slice thickness: 1 mm) at 16 different diffusion weightings, i.e., b-values (0, 68, 85, 117, 132, 189, 332, 520, 615, 709, 803, 897, 993, 1093, 1193, 1492, and 1791 s/mm^2^). The b-values were chosen to homogenously spread a range of b-values capturing, on one end, the non-linear signal decrease at low b-values (for the calculation of IVIM-related parameters *D** and *fIVIM*) and, on the other end, the linear signal decrease at intermediate diffusion weightings (for the calculation of the diffusion coefficient *D*).

### 2.4. Tumor Resection and Fluorescent mpIHC

Following excision, to maintain the exact anatomical position of breast cancer tumors, the left, right, and dorsal sides of each breast cancer tumor were marked using different-colored tissue dyes (Epredia™ Shandon™ Tissue-Marking Dyes, Fisher Scientific, Hampton, NH, USA). Then, the breast cancer tumors were cut in halves matching the same axial cutting plane that was measured on mpMRI. Sliced breast cancer tumors were then fixed in 4% formaldehyde (ROTI^®^ Histofix, Roth, Karlsruhe, Germany) for 24 h before being placed in 70% EtOH for storage at 4 °C. Subsequently, the fixed samples were paraffin-embedded and cut in 3 µm slice thickness. Fluorescent mpIHC was performed by CBmed, Graz, Austria. The slices were stained for CD31 PDGFR-β and Hif1-α. CD-31 is an established staining target of vessel-forming endothelial cells that has previously been used as an ex vivo biomarker for microvessel density in breast cancer [38,39]. PDGFR-β plays a pivotal role in the maturation of blood vessels: endothelial cells presenting PDGFR-like receptors are involved in the recruitment of pericytes, increasing pericyte density at newly formed blood vessels, which increases blood-vessel contractility to allow continuous blood flow [40,41,42,43,44]. Lastly, Hif1-α was included in the staining protocol as a direct ex vivo biomarker of hypoxia [45,46]. 

### 2.5. Imaging Post-Processing

#### 2.5.1. BOLD MRI

A detailed description of the results for BOLD MRI in this sample of mice has been published previously by our group [27]. BOLD parameter maps for baseline (21% O_2_) and challenged (100% O_2_) calculations were generated using MATLAB code (MATLAB version R2018a) developed in-house. Firstly, parameter maps for the *T*_2_*** relaxation rate, *R*_2_***, were calculated as the inverse of *T*_2_***. Secondly, average parameter maps (${{{R_{2}}^{*}}_{avg}}_{\left(baseline\right)}$) over three baseline scans were generated. Thirdly, the voxel-wise difference between the challenged and the averaged baseline scans was calculated for each challenged measurement cycle separately using the following equation:(1)$$∆{R_{2}}^{*}={{R_{2}}^{*}}_{\left(100\right)}-{{{R_{2}}^{*}}_{avg}}_{\left(baseline\right)}$$
where ${{R_{2}}^{*}}_{\left(100\right)}$ corresponds to the challenged BOLD measurement at 100% oxygen, while ${{{R_{2}}^{*}}_{avg}}_{\left(baseline\right)}$ corresponds to the averaged baseline measurement.

Lastly, in order to discriminate responsive from non-responsive voxels, ${{∆R}_{2}}^{*}$ parameter maps were filtered according to the following equation proposed by Yang et al. 2020:(2)$$|{{∆R}_{2}}^{*}|\left\{\begin{array}{c} <2*SD\left({{{∆R}_{2}}^{*}}_{21\%}\right):\;non-responsive\\ >2*SD\left({{{∆R}_{2}}^{*}}_{21\%}\right):\;responsive \end{array}\right.$$
where $SD\left({{{∆R}_{2}}^{*}}_{21\%}\right)$ refers to the standard deviation of *R*_2_*** in each voxel over the three measurement cycles during baseline image acquisition. Following these calculations, only responsive voxels were included in the subsequent analyses, which were performed using the ITK-SNAP (version 3.6.0) software. Regions of interest (ROIs) were drawn on the slice including the largest tumor diameter and the least artifacts from breathing motion, while avoiding obvious necrotic tumor regions identified on the corresponding slice in anatomical reference images. 

#### 2.5.2. IVIM MRI

Bruker raw image datasets of IVIM-MRI acquisitions were converted into an FMRIB Software Library (FSL) dataset using the MRIcroGL (version 1.2.20210317) tool. DWI data were analyzed using the MITKdiffusion (version 1.2.0, German Cancer Research Center, Heidelberg, Germany) tool. To generate IVIM-MRI parameter maps, data were fitted using a segmented fitting approach: All signal values above a b-value of 350 s/mm^2^ were used in an initial fit to obtain parameter maps for the diffusion coefficient *D*, as well as the IVIM fraction *fIVIM*. Next, signal values below 350 s/mm^2^ were fitted to obtain the perfusion coefficient *D**. Then, two-dimensional ROIs were placed onto the parameter maps in the center of the tumor, excluding cystic and necrotic areas identified on reference T1-weighted anatomical reference images. The selection of the respective slice was based on the slice selection on the BOLD-MRI images. For visualization, IVIM- and BOLD-MRI parameter maps were coregistered with the T2-weighted anatomical reference image using the pmod (version 4.3) tool.

#### 2.5.3. Fluorescent mpIHC

Histological sections that were multiplex-stained for CD31, PDGFR-β, and Hif1-α were analyzed using the Halo module HighPlex FL (version 3.2.1, Indica Labs, Albuquerque, NM, USA). The total number of cells and the number of stained cells were quantified to calculate the fraction of positively stained cells for each histological marker.

### 2.6. Statistical Analysis

All statistical analyses were performed using the RStudio (version 1.2.5033) software. To assess differences among breast-cancer molecular subtypes for each in vivo and ex vivo imaging parameter, Kruskal–Wallis inferential testing was applied, followed by Benjamini–Hochberg post hoc *p*-value adjustment. Comparisons with a *p*-value < 0.05 were considered statistically significant. Multiparametric analysis of histological and MRI parameters was conducted using principal-component analysis (PCA).

## 3. Results

### 3.1. IVIM-MRI and BOLD-MRI Parameter Maps

Breast cancer xenograft tumors grew to an average maximal diameter of 7.1 ± 1.9 mm in luminal A breast cancers, 7.8 ± 2.3 mm in Her2+ breast cancers, and 7.9 ± 1.6 mm in triple-negative breast cancers.

An exemplary illustration of IVIM-MRI and BOLD-MRI parameter maps for a luminal A breast cancer xenograft tumor, as well as the corresponding hematoxylin-and-eosin (H&E)-stained histological section, are shown in Figure 1. Note that the IVIM-MRI parameter maps were prone to distortion because of the EPI sequence used for their measurement. The BOLD-MRI parameter map only shows pixels that were identified as responsive to the hyperoxic gas challenge according to the classification described above.

### 3.2. mpMRI

Table 1 presents the summary statistics of BOLD-MRI and IVIM-MRI parameters (all values in median, interquartile range) across breast-cancer molecular subtypes. 

#### 3.2.1. BOLD-MRI

The BOLD-MRI-related parameter Δ*R2**, a surrogate marker for oxygen delivery via the blood, exhibited significant differences between luminal A and triple-negative breast cancers, as well as between Her2+ and triple-negative breast cancers (both *p* < 0.001; see Table 1). Figure 2d illustrates the differences in Δ*R2** between breast-cancer molecular subtypes, incorporating data from all three BOLD-MRI measurement cycles in the form of a boxplot.

#### 3.2.2. IVIM MRI

Differences in *fIVIM*, a surrogate marker for the fraction of microvessel perfusion within a respective voxel, were statistically significant (*p* = 0.04) between Her2+ and triple-negative breast cancers. Additionally, the comparison of *fIVIM* between luminal A and triple-negative breast cancers approached statistical significance (*p* = 0.05). However, the comparison of *D*, representing extravascular diffusion, and *D**, representing intravascular perfusion, did not reveal any significant differences among the breast-cancer molecular subtypes. Figure 2a–c illustrate the differences of the IVIM-MRI-related parameters *D*, *fIVIM,* and *D** between breast-cancer molecular subtypes.

#### 3.2.3. Multivariate Analysis

The outcomes of PCA integrating all mpMRI parameters (*D*, *fIVIM*, *D**, and Δ*R_2_**) in this study are illustrated in Figure 3. PCA revealed a distinct segregation between triple-negative and luminal A breast cancers and between Her2+ and triple-negative molecular breast cancers, primarily along the first principal component (PC1). The segregation was influenced predominantly by variations in the parameters *fIVIM* and Δ*R2**, both of which contributed significantly to PC1 in terms of absolute values. Of note, for the comparison shown in Figure 3, Δ*R2** measurements from the first BOLD measurement cycle (of three BOLD measurement cycles) were used. Table 2 gives a detailed overview on the loadings of each principal component. 

### 3.3. Fluorescent mpIHC

Table 3 details summary statistics (median, interquartile range) of the fraction of cytoplasms stained for CD31, PDGFR-β, and Hif1-α. Statistically significant differences in CD31 staining were observed between luminal A and Her2+ breast cancers (*p* < 0.05), as well as between luminal A and triple-negative breast cancers (*p* < 0.005, Figure 4).

The outcomes of PCA integrating CD31, PDGFR-β, and Hif1-α parameters derived from ex vivo fluorescent mpIHC are summarized in Figure 5. Luminal A breast cancers were differentiated from Her2+ and triple-negative subtypes primarily along an axis defined by the first and second principal components. Clusters were mainly characterized by differences in the expression levels of CD31 and PDGFR-β. Table 4 gives a detailed overview on the contributions of each target protein to the principal components.

## 4. Discussion

Our study shows that mpMRI of hypoxia and hypoxia-induced neoangiogenesis without the application of GBCA allows clear separation of triple-negative breast cancers from other breast-cancer molecular subtypes. The results are complemented by a multivariate analysis of mpIHC-based biomarkers for hypoxia and hypoxia-induced neoangiogenesis, in which luminal A breast cancers were clearly separated from Her2+ and triple-negative breast cancers.

To our knowledge, this preclinical pilot study is the first to highlight that breast cancers of different molecular subtypes can be differentiated based on differences in hypoxia and hypoxia-induced neoangiogenesis derived from non-invasive, non-contrast-enhanced mpMRI. Of note, despite the high sensitivity of dynamic contrast-enhanced (DCE) MRI for breast-cancer detection, its utility for molecular subtyping remains limited. In addition, the use of GBCA has recently been questioned over the associated costs and logistics, risk for hypersensitivity reactions, systemic nephrogenic fibrosis, and potential for accumulation in the brain [47]. Of note, no clinical symptoms have been associated with the deposition of macrocyclic GBCA [48,49]. Nevertheless, the identification of new imaging markers that can exploit endogenous contrast instead of relying on GBCA for improved breast-cancer characterization is desirable. Non-contrast-enhanced alternatives continue to be explored, exemplified most successfully by DWI with apparent diffusion coefficient mapping, which provides a surrogate marker of increased tissue density in cancers [33,34,50]. The addition of DWI to standard DCE-MRI protocols has been shown to increase the specificity of breast-cancer molecular subtyping as well as of the prediction of treatment response, therefore demonstrating the potential of non-contrast-enhanced mpMRI-based imaging biomarkers over DCE-MRI imaging biomarkers alone [23,50].

Technological improvements in hardware and software have spurred investigations of non-contrast-enhanced imaging biomarkers beyond those of DWI that are highly sensitive to the physiological changes involved in breast-cancer progression and in the development of aggressive breast-cancer molecular subtypes. For instance, extending routine DWI measurements to include low b-values enables the quantification of the microvasculature-specific IVIM effect. The IVIM-MRI-related parameters *D** and *fIVIM* have previously been explored in breast cancer xenografts as well as in clinical studies [35,36,51,52,53,54]. Our study indicates that *fIVIM* can be used to differentiate triple-negative from Her2+, as well as luminal A, breast cancers. Triple-negative breast cancers have the lowest *fIVIM* values among the investigated breast-cancer molecular subtypes in our study, reflecting their compromised microvascular blood flow. In agreement with previous studies that found that the quantification of *D** is particularly challenging, we did not find the parameter *D* useful for the characterization of molecular subtypes of breast cancers, although it may be useful for the discrimination of malignant and benign breast lesions in a clinical setting [53]. 

Another non-contrast-enhanced MRI parameter explored in this study is the BOLD-MRI-related Δ*R2** parameter, which utilizes blood as an endogenous contrast agent. This parameter depends on the oxygenation of erythrocyte-bound hemoglobin, creating local susceptibility gradients detectable as MRI contrast. The decrease in *R2** following a hyperoxic gas challenge, indicative of increased oxygenated blood flow, was most pronounced in luminal A breast cancers, as previously demonstrated by our group [27].

In this preliminary preclinical study, we explored novel non-invasive non-contrast-enhanced MRI markers for hypoxia and hypoxia-induced angiogenesis. Clear distinctions were observed between luminal A and triple-negative breast cancers, while there was overlap between intermediate Her2+ and the luminal A breast cancers. These differences were mainly rooted in the significant differences between breast-cancer molecular subtypes in the IVIM-MRI-related parameter *fIVIM* and the BOLD-MRI-related parameter Δ*R*_2_***, as shown by the parameters’ loadings on the first principal component. Unlike single-parametric analyses, PCA-based clustering of breast cancer xenograft data revealed no overlap between luminal A and triple-negative breast cancers. These findings suggest that this novel non-contrast-enhanced mpMRI approach integrating both IVIM MRI and BOLD MRI may enhance the specificity of breast-cancer molecular subtyping in clinical investigations and allow a more comprehensive characterization of the heterogeneous hypoxic tumor microenvironment [21,54].

Furthermore, we compared mpMRI analyses with mpIHC stainings of breast cancer xenografts to validate the performance of the imaging parameters. The mpIHC approach enabled the assessment of multiple markers (CD31, PDGFR-β, and Hif1-α) on the same tissue slice as that of the imaging slice. While CD31 levels were higher in Her2+ and triple-negative cancers, indicating increased microvessel density, the low PDGFR-β presence suggested limited blood-vessel maturation. In contrast, luminal A cancers exhibited fewer CD31-positive cells and higher PDGFR-β levels, implying more efficient oxygen delivery. However, differences in Hif1-α expression were not statistically significant between breast-cancer molecular subtypes. Additionally, multivariate analysis of mpIHC data showed clear separation between luminal A and triple-negative breast cancers, with Her2+ breast cancers largely overlapping with triple-negative breast cancers. These differences were driven by differences in CD31 and PDGFR-β expression. Overall, the findings from mpIHC analysis corroborate those from mpMRI, indicating that while triple-negative cancers may have higher microvessel density, their vascular maturity is insufficient for effective oxygen delivery. In contrast, the higher *fIVIM* and more pronounced Δ*R2** in luminal A breast cancers suggest better oxygen delivery compared to triple-negative breast cancers.

In regard to the limitations in this study, corresponding mpIHC data were not available for all imaging data, and therefore, insights into the correlations between histological and MRI parameters are limited. In addition, our study relied on data from xenograft tumors only. For a more in-depth assessment of tumor hypoxia, longitudinal studies that capture the dynamic changes in the hypoxic tumor microenvironment and include inter-observer reliability assessment should be performed, which was beyond the scope of the currently presented project. Future translational studies should investigate further whether a non-contrast-enhanced mpMRI approach using hyperoxic BOLD MRI and IVIM MRI is also useful for the discrimination of breast cancers in patients. 

## 5. Conclusions

A non-contrast-enhanced mpMRI protocol combing hyperoxic-BOLD-MRI and IVIM-MRI measurements was shown to allow the discrimination of triple-negative from luminal A and Her2+ breast cancers and the assessment of hypoxia and hypoxia-induced angiogenesis in xenograft models of breast cancer, with mpIHC data supporting mpMRI findings. Larger-scale preclinical and translational future studies should investigate correlations between mpIHC-based and mpMRI-based biomarkers of hypoxia and hypoxia-induced angiogenesis.

## Figures and Tables

**Figure 1 cancers-16-00375-f001:**
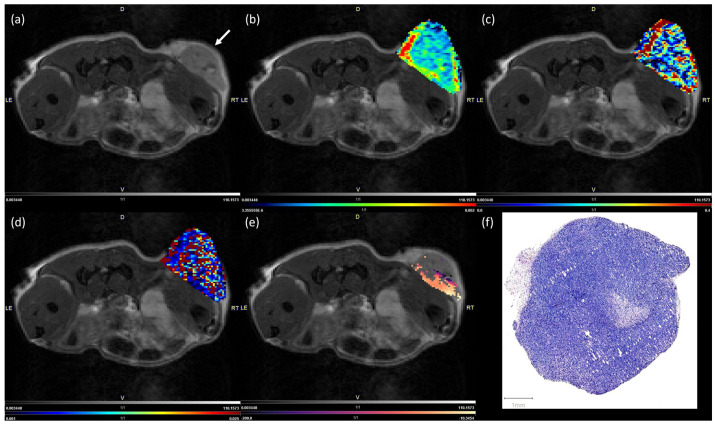
Exemplary parameter maps and a hematoxylin-and-eosin (H&E)-stained tissue section of a luminal A breast cancer xenograft tumor. T2-weighted image used for anatomical reference. The white arrow indicates the location of the tumor (**a**). Parameter maps of the IVIM-MRI-derived diffusion coefficient *D* (**b**), the IVIM-MRI fraction *fIVIM* (**c**), and the perfusion coefficient *D** (**d**), masked onto the anatomical reference image. The BOLD-MRI-related parameter map for Δ*R2** during the first of three measurement cycles, masked onto the anatomical reference image (**e**). An H&E-stained tissue section of the tumor (**f**).

**Figure 2 cancers-16-00375-f002:**
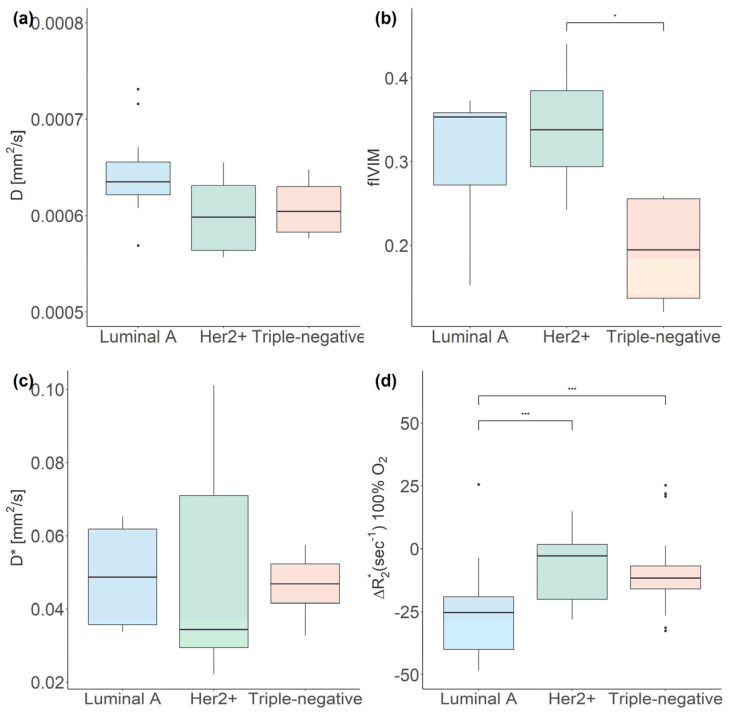
Boxplots illustrating the median and interquartile range of the IVIM-MRI-related parameters *D* (**a**), *fIVIM* (**b**), and *D** (**c**), as well as the BOLD-MRI-related parameter Δ*R_2_** (**d**), across breast-cancer molecular subtypes. (* *p* < 0.05; *** *p* < 0.001).

**Figure 3 cancers-16-00375-f003:**
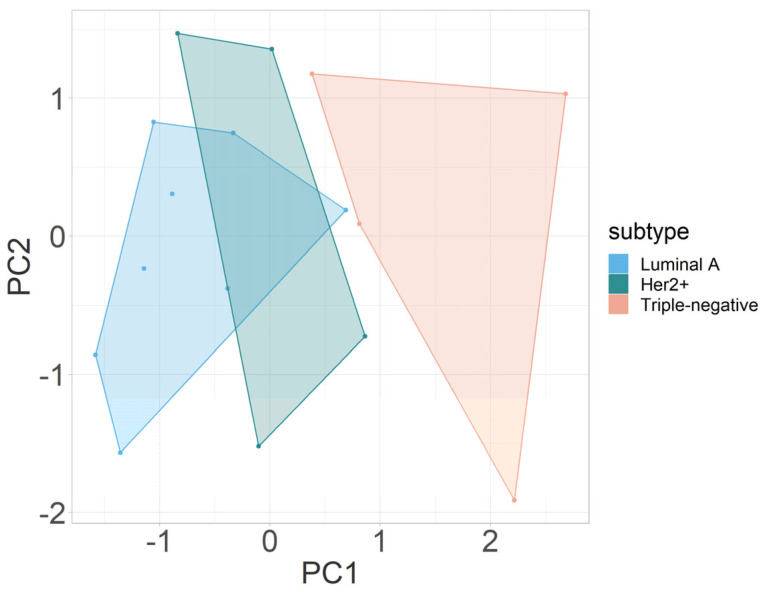
Principal-component analysis of multiparametric MRI parameters *D*, *fIVIM*, *D**, and Δ*R*_2_* to distinguish between luminal A, Her2+, and triple-negative breast cancers. Abbreviations: PC, principal component.

**Figure 4 cancers-16-00375-f004:**
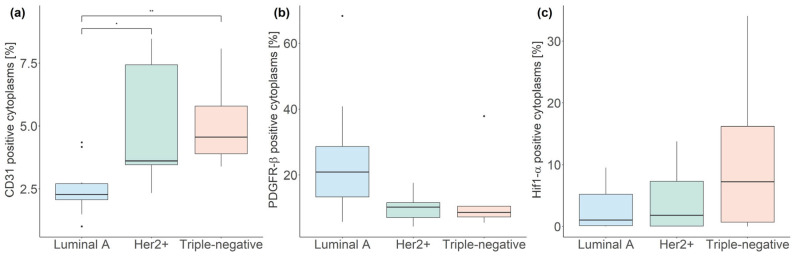
Boxplots illustrating the median and interquartile range of multiplexed immunohistochemistry stains for CD31 (**a**), PDGFR-β (**b**), and Hif1-α (**c**) across breast-cancer molecular subtypes. (* *p* < 0.05; ** *p* < 0.005).

**Figure 5 cancers-16-00375-f005:**
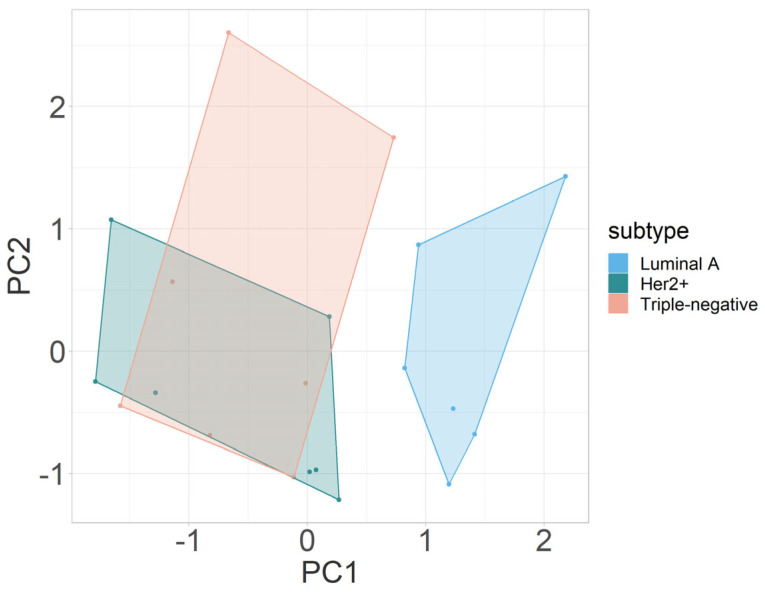
Principal-component analysis of multiplexed immunohistochemistry parameters CD31, PDGFR-β, and Hif1-α. Abbreviations: PC, principal component.

**Table 1 cancers-16-00375-t001:** Summary statistics (median, interquartile range) for multiparametric MRI (BOLD-MRI and IVIM-MRI) parameters (*D*, *fIVIM*, *D**, and Δ*R_2_**) for luminal A, Her2+, and triple-negative breast cancers.

Subtype	IVIM MRI			BOLD MRI
	*D* (10^−3^ mm^2^/s)	*fIVIM* (%)	*D** (mm^2^/s)	Δ*R_2_**
Luminal A	0.6350, 0.034	35.3, 8.6	0.0487, 0.026	−28.42, 24.35
Her2+	0.5985, 0.00673	33.8, 9.1 ^a^	0.0343, 0.042	−2.85, 21.95 †
Triple-negative	0.6240, 0.0063	19.5, 11.9	0.0468, 0.011	−11.64, 9.15 †

^a^ Within a column, superscripts indicate statistically significant differences from triple-negative breast cancers (*p* < 0.05). † within a column, a dagger indicates statistically significant differences from luminal A breast cancers (*p* < 0.001).

**Table 2 cancers-16-00375-t002:** Loadings for principal-component analysis on the multiparametric MRI parameters *D*, *fIVIM*, *D**, and Δ*R*_2_* to distinguish between luminal A, Her2+, and triple-negative breast cancers.

Parameter	PC1	PC2	PC3	PC4
*fIVIM*	0.6716729	−0.3464966	−0.2224006	0.6159007
*D**	−0.1683178	−0.7922028	−0.4168883	−0.4126595
*D*	−0.4323694	−0.4692890	0.6342436	0.4365314
Δ*R*_2_***	−0.5775650	0.1792278	−0.6119454	0.5097243

Abbreviations: PC, principal component.

**Table 3 cancers-16-00375-t003:** Summary statistics (median, interquartile range) for fluorescent multiplexed immunohistochemistry stains of CD31, PDGFR-β, and Hif1-α for luminal A, Her2+, and triple-negative breast cancers.

Subtype	*CD*31 (%)	*PDGFR-β* (%)	*Hif*1*-α* (%)
Luminal A	2.27, 0.64	20.89, 15.29	1.04, 5.07
Her2+	3.61, 3.97 *	10.25, 4.51	1.82, 7.28
Triple-negative	4.56, 1.89 **	8.59, 3.24	7.26, 15.51

* Within a column, asterisks indicate statistically significant differences from luminal A breast cancers (* *p* < 0.05; ** *p* < 0.005).

**Table 4 cancers-16-00375-t004:** Loadings for principal-component analysis on the multiplexed immunohistochemistry parameters CD31, PDGFR-β, and Hif1-α to distinguish between luminal A, Her2+, and triple-negative breast cancers.

Parameter	PC1	PC2	PC3
PDGFR-β	0.6791799	0.410240	0.608620
Hif1-α	−0.0863886	0.868131	−0.488759
CD31	−0.7288701	0.279378	0.625057

Abbreviations: PC, principal component.

## Data Availability

The data presented in this study are available on request from the corresponding author.

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
