# Peer review of "Non-Contrast-Enhanced Multiparametric MRI of the Hypoxic Tumor Microenvironment Allows Molecular Subtyping of Breast Cancer: A Pilot Study"

_cancers, 2024, doi:10.3390/cancers16020375_

Round 1

Reviewer 1 Report

Comments and Suggestions for Authors

The manuscript could benefit from more detailed explanations of the MRI techniques used, particularly the specifics of BOLD-MRI and IVIM-MRI. This would aid in replicability and understanding of the study. 

If applicable, conduct a dose-response experiment to understand the relationship between the intensity or duration of MRI exposure and the observed effects on the tumor microenvironment.

To address the subjectivity in interpreting MRI results, an inter-observer variability study can be conducted. Different radiologists or technicians would independently analyze the same MRI images to assess the consistency of the interpretations.

A longitudinal study could be performed to observe changes in the tumor microenvironment over time using the MRI techniques. This would provide insights into the temporal dynamics of the tumor's hypoxic state.

Author Response

We thank Reviewer 1 for his valuable comments, which greatly improved the manuscript.

Comment 1: The manuscript could benefit from more detailed explanations of the MRI techniques used, particularly the specifics of BOLD-MRI and IVIM-MRI. This would aid in replicability and understanding of the study. 

Answer 1: Along with the changes to the explanation to BOLD-MRI, we have added a more detailed explanation of the IVIM-MRI technique in the Introduction of the manuscript:

            Lines 92ff: “Routinely used DWI protocols, which include b-values between 300 s/mm2 and 1000 s/mm2, enable a quantification of the apparent diffusion coefficient, a surrogate marker of tissue cellularity, via the signal decrease due to random proton diffusion, Following the measurement of b-values lower than 300 s/mm2 in IVIM-MRI protocols, the non-random, microvascular diffusion contributes predominantly to the signal loss, and the perfusion-related IVIM fraction fIVIM and perfusion coefficient D* can be estimated.”

Comment 2: If applicable, conduct a dose-response experiment to understand the relationship between the intensity or duration of MRI exposure and the observed effects on the tumor microenvironment.

Answer 2: Due to the lack of ionizing radiation in MRI examinations, no dose-dependent effects on the tumor microenvironment are to be expected.

Comment 3: To address the subjectivity in interpreting MRI results, an inter-observer variability study can be conducted. Different radiologists or technicians would independently analyze the same MRI images to assess the consistency of the interpretations.

Answer 3: An inter-observer reliability study was beyond the scope of the presented pilot project. Contrary to clinical studies, the tumor margins are easily detected in preclinical data (see, e.g., Figure 1 of the manuscript). We are confident that the planned large-scale study, which will also include inter-observer reliability analyses, will corroborate the findings. We have included the absence of inter-observer reliability analyses in the limitations-section of the Discussion:

Lines 443ff: "For a more in-depth assessment of tumor hypoxia, longitudinal studies which capture the dynamic changes in the hypoxic tumor microenvironment and include inter-observer reliability assessment should be performed, which was beyond the scope of the currently presented project."

Comment 4: A longitudinal study could be performed to observe changes in the tumor microenvironment over time using the MRI techniques. This would provide insights into the temporal dynamics of the tumor's hypoxic state.

Answer 4: We appreciate the importance of longitudinal studies to track changes in the hypoxic tumor microenvironment. We have highlighted this issue in the limitations-section of the Discussion:

            Lines 443ff: “For a more in-depth assessment of tumor hypoxia, longitudinal studies which capture the dynamic changes in the hypoxic tumor microenvironment and include inter-observer reliability assessment should be performed, which was beyond the scope of the currently presented project."

Reviewer 2 Report

Comments and Suggestions for Authors

This well-written paper presents new and promising results concerning a specific area of breast oncology where MRI may become particularly useful for prognostication and therapy decision-making.

Please rephrase the third sentence in Discussion, (p10, l344-346) so as not to make it seem you are trying to overemphasize the risks associated with IV application of gadolinium-based contrast agents. I am specifically referring to nephrogenic systemic fibrosis and gadolinium deposition in brain, both of which are associated with linear chelates, and with the latter having no association with any clinical symptoms almost 10 years following the first report (also see references below).

1.      Weinreb JC, Rodby RA, Yee J, Wang CL, Fine D, McDonald RJ, Perazella MA, Dillman JR, Davenport MS. Use of Intravenous Gadolinium-based Contrast Media in Patients with Kidney Disease: Consensus Statements from the American College of Radiology and the National Kidney Foundation. Radiology. 2021 Jan;298(1):28-35. doi: 10.1148/radiol.2020202903. Epub 2020 Nov 10. PMID: 33170103.

2.      van der Molen AJ, Quattrocchi CC, Mallio CA, Dekkers IA; European Society of Magnetic Resonance in Medicine, Biology Gadolinium Research, Educational Committee (ESMRMB-GREC). Ten years of gadolinium retention and deposition: ESMRMB-GREC looks backward and forward. Eur Radiol. 2023 Oct 7. doi: 10.1007/s00330-023-10281-3. Epub ahead of print. Erratum in: Eur Radiol. 2023 Nov 15;: PMID: 37804341.

Author Response

We thank Reviewer 2 for his valuable comments, which greatly improved the manuscript.

Comment 1: Please rephrase the third sentence in Discussion, (p10, l344-346) so as not to make it seem you are trying to overemphasize the risks associated with IV application of gadolinium-based contrast agents. I am specifically referring to nephrogenic systemic fibrosis and gadolinium deposition in brain, both of which are associated with linear chelates, and with the latter having no association with any clinical symptoms almost 10 years following the first report (also see references below).

  1. Weinreb JC, Rodby RA, Yee J, Wang CL, Fine D, McDonald RJ, Perazella MA, Dillman JR, Davenport MS. Use of Intravenous Gadolinium-based Contrast Media in Patients with Kidney Disease: Consensus Statements from the American College of Radiology and the National Kidney Foundation. Radiology. 2021 Jan;298(1):28-35. doi: 10.1148/radiol.2020202903. Epub 2020 Nov 10. PMID: 33170103.

  1. van der Molen AJ, Quattrocchi CC, Mallio CA, Dekkers IA; European Society of Magnetic Resonance in Medicine, Biology Gadolinium Research, Educational Committee (ESMRMB-GREC). Ten years of gadolinium retention and deposition: ESMRMB-GREC looks backward and forward. Eur Radiol. 2023 Oct 7. doi: 10.1007/s00330-023-10281-3. Epub ahead of print. Erratum in: Eur Radiol. 2023 Nov 15;: PMID: 37804341.

Answer 1: We have rephrased the respective paragraph in the Discussion:

                Lines 376 ff: “In addition, the use of GBCA has recently been questioned over the associated costs and logistics, risk for hypersensitivity reactions, systemic nephrogenic fibrosis, and potential for accumulation in the brain [47]. Of note, no clinical symptoms have been associated with the deposition of macrocyclic GBCA [48, 49].”

Reviewer 3 Report

Comments and Suggestions for Authors

The paper is suitable for publication in Cancers. I suggest a few clarifications:

1. in Introduction should you mention the well-known term functional MRI?

2. in Figure 3 and 5 colors for LumA and HER2 are almost indistinguishable.

Author Response

We thank Reviewer 3 for his valuable comments, which greatly improved the quality of the manuscript.

Comment 1: in Introduction should you mention the well-known term functional MRI?

Answer 1: We have included an explanation of the term functional MRI in the context of the BOLD-effect in the Introduction:

            Lines 84ff: “The dependency of BOLD-MRI signal on blood oxygenation is most frequently associated with brain functional MRI studies, where a signal change is associated with regional brain activity.”

Comment 2: in Figure 3 and 5 colors for LumA and HER2 are almost indistinguishable.

Answer 2: We have adjusted Figure 3 and Figure 5 with more intense color tones for better readability.

Round 2

Reviewer 1 Report

Comments and Suggestions for Authors

This manuscript is now significantly improved and I am affirmative to publish this.